# Assessment of three antibiotic combination regimens against Gram-negative bacteria causing neonatal sepsis in low- and middle-income countries

Gram-negative bacteria (GNB) are a major cause of neonatal sepsis in low- and middle-income countries (LMICs). Although the World Health Organization (WHO) reports that over 80% of these sepsis deaths could be prevented through improved treatment, the efficacy of the currently recommended first- and second-line treatment regimens for this condition is increasingly affected by high rates of drug resistance. Here we assess three well known antibiotics, fosfomycin, flomoxef and amikacin, in combination as potential antibiotic treatment regimens by investigating the drug resistance and genetic profiles of commonly isolated GNB causing neonatal sepsis in LMICs. The five most prevalent bacterial isolates in the NeoOBS study (NCT03721302) are *Klebsiella pneumoniae, Acinetobacter baumannii, E. coli, Serratia marcescens* and *Enterobacter cloacae* complex. Among these isolates, high levels of ESBL and carbapenemase encoding genes are detected along with resistance to ampicillin, gentamicin and cefotaxime, the current WHO recommended empiric regimens. The three new combinations show excellent in vitro activity against ESBL-producing *K. pneumoniae* and *E. coli* isolates. Our data should further inform and support the clinical evaluation of these three antibiotic combinations for the treatment of neonatal sepsis in areas with high rates of multidrug-resistant Gram-negative bacteria.

In 2019, more than 560,000 neonatal deaths were associated with bacterial antimicrobial resistance (AMR), including nearly 140,000 deaths directly attributable to bacterial AMR[1]. The World Health Organization (WHO) reports that over 80% of sepsis deaths could be prevented if there was improved treatment and infection prevention[2].

Most of these cases occur in low- and middle-income countries (LMICs)[3]. Of the multiple large studies recently conducted in these countries[1,4–6], one systematic review[7] and several single-site reports[8–10] have shown that Gram-negative bacteria (GNB), such as *Klebsiella* spp., *Escherichia coli*, and *Acinetobacter baumannii*, are considered the main cause of neonatal sepsis in approximately 40% of episodes.

Furthermore, these studies have demonstrated that the empiric treatment of neonatal sepsis currently recommended by WHO, which includes a narrow-spectrum β-lactam antibiotic in combination with gentamicin as a first line regimen and a 3rd generation cephalosporin as a second line regimen[11], is increasingly compromised by high drug resistance rates, particularly due to the high prevalence of ESBLs and of aminoglycoside modifying enzymes (AMEs). Findings from these recent studies extend data from several previous reports from LMICs where extremely high rates of resistance to amoxicillin (80%), gentamicin (60%), and third-generation cephalosporins (>80%) were observed[12–17].

✉ e-mail: Biljana.KakaraskoskaBoceska@uantwerpen.be; gyglupczynski@gmail.com

The NeoOBS study (NCT03721302)[6,18] was a prospective, multi-center, observational cohort study investigating the management of neonatal sepsis in several countries, aiming to inform and enhance the design of the current and future studies investigating new potential antibiotic treatments.

Three generic antibiotics, amikacin, flomoxef and fosfomycin were selected that met the criteria for consideration in the trial[19], and their potential to be used in novel combined empirical regimens was assessed by a dynamic hollow-fiber infection model (HFIM) and by checkerboard assays[20–22]. The three combinations, fosfomycin–amikacin, fosfomycin–flomoxef and flomoxef–amikacin exhibited synergistic interactions measured by both bactericidal killing and the prevention of emergence of resistance.

The NeoOBS microbiology sub-study aimed to determine the level of antibiotic susceptibility of multidrug-resistant (MDR) Gram-negative bacteria to currently used antibiotics and also to assess the novel drug combinations included in the NeoSep1 antibiotic trial (ISRCTN48721236).

## Results

In total, 420 Gram-negative bacterial isolates were received. The five most common identified species were: *K. pneumoniae* (*n* = 135), *A. baumannii* (*n* = 80), *E. coli* (*n* = 34), *S. marcescens* (*n* = 33) and *Enterobacter cloacae* complex (ECC) (*n* = 27). These 309 Gram-negative isolates were obtained from 295 patients. The remaining 111 GNB isolates belonged to 18 different bacterial species and were represented by less than 20 isolates each. The distribution of the five major GNB isolates per site is shown in Fig. 1.

### Antibiotic susceptibility testing and resistance mechanisms

Ampicillin/gentamicin regimen had low rates of coverage, with 32% (34/108) of the isolates susceptible, and only 22% (24/108) of them were susceptible to cefotaxime (Supplementary Table 1, Supplementary Data 1).

*K. pneumoniae* was the species with highest rates of resistance due to the high prevalence of ESBLs and of AMEs, found in 74% (64/87) and 63% (55/87) of the isolates, respectively. The rates of susceptibility to piperacillin-tazobactam differed between *K. pneumoniae* (35%) and *E. coli* (67%) due to the frequent occurrence of the *bla*<sub>OXA-1</sub> gene among *K. pneumoniae* isolates. Carbapenem resistance was almost exclusive to *K. pneumoniae*, with 30% (26/87) strains resistant to meropenem versus 5% (1/21) of *E. coli*. All of these strains were associated with the presence of carbapenemase encoding genes, with the most prevalent for *K. pneumoniae* being NDM-like (81%; *n* = 21/26), and OXA-48-like (11%; *n* = 3/26).

Susceptibility rates to flomoxef (71%) and amikacin (70%) were comparable to those observed for meropenem (75%) in the subset of 108 *E. coli* and *K. pneumoniae* isolates with available MIC results. Flomoxef displayed excellent activity against ESBL-producing isolates but resistance to this agent was only due to class C β-lactamase (AmpC cephalosporinase) or to carbapenemase-producing strains. Among the aminoglycosides, amikacin showed the best activity since it was not affected by common AMEs (AAC(3)-II, ANT(2")-I), which are known to modify gentamicin and tobramycin. Fosfomycin showed the strongest antibacterial activity with 90% of the tested isolates being susceptible to this antibiotic. An almost perfect match (106/108) was found between susceptibility and resistance phenotypes and genotypes for

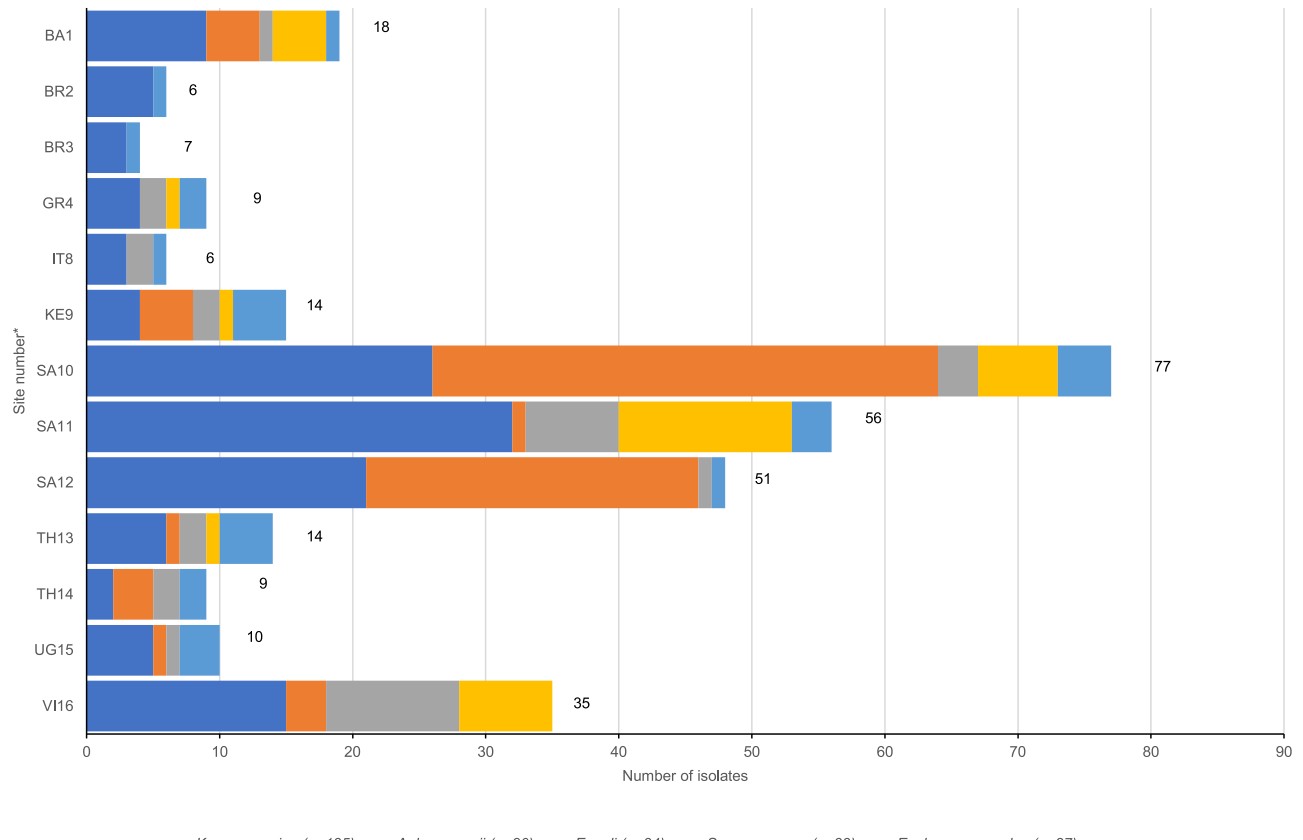

**Fig. 1 | Distribution of the total number (indicated next to the bars) of the five most common GNB species analyzed by site.** *n* = 309, one isolate per species per patient only, following removal of duplicates. Number of isolates correlates with the number of neonates, except for the following sites: TH13 (13 neonates–14 isolates), SA12 (46 neonates–51 isolates), SA11 (51 neonates–56 isolates), SA10 (75 neonates–77 isolates) and BR2 (5 neonates–6 isolates).* VI Vietnam, UG Uganda, TH Thailand, SA South Africa, KE Kenya, IT Italy, GR Greece, BR Brazil, BA Bangladesh. The numbers following the country keys refer to the site number (see Table 1). Source data are provided as a Source Data file.

**Table 1 | Susceptibility of 108 _K. pneumoniae_ and _E. coli_ isolates to different combination regimens based on the novel combination breakpoint thresholds[20–22]**

| _K. pneumoniae_ and _E. coli_* isolates tested for MIC determination | FOS/FLX | FOS/AMK | FLX/AMK |
|---|---|---|---|
| | **N (% coverage)** | | |
| Cefotaxime-S (n = 24) | 24 (100) | 23 (96) | 24 (100) |
| _ESBL_-negative | | | |
| Cefotaxime-R/meropenem-S (n = 58) | 55 (95) | 52 (90) | 53 (91) |
| ESBL-positive/Carbapenemase-negative (n = 52) | 52 (100) | 48 (92) | 52 (100) |
| AmpC-positive/Carbapenemase-negative (n = 6) | 3 (50) | 4 (67) | 1 (17) |
| Meropenem-R (n = 26) | 3 (12) | 10 (38) | 0 (0) |
| Carbapenemase-positive | | | |
| Total number of isolates (n = 108) | 82 (76) | 85 (79) | 77 (71) |

*_Klebsiella pneumoniae_ (n = 87), _Escherichia coli_ (n = 21). FOS fosfomycin, FLX flomoxef, AMK amikacin.

all beta-lactams, including broad-spectrum penicillins, cephalosporins, and carbapenem drugs, as well as for aminoglycosides. All ESBLs, AmpC and carbapenem resistance genes detected by genotyping correctly predicted strain susceptibility patterns, except for one _K. pneumoniae_ isolate carrying the $bla_{NDM-1}$ gene but that was found to be susceptible to meropenem (MIC of 0.06 mg/L). For aminoglycosides, all strains carrying genes encoding for enzymes modifying gentamicin and/or to amikacin (AAC(6′)-I, APH(3′)-VI) were confirmed as resistant to these drugs by MIC testing. All strains carrying genes encoding for 16 S rRNA methylases displayed high level resistance to gentamicin and to amikacin (MIC > 256 mg/L for both agents). Among the 11 fosfomycin-resistant strains, only two carried the gene encoding the fosfomycin-modifying enzyme (_fosA3_ in one _E. coli_ isolate with a fosfomycin of MIC > 512 mg/L and _fosA5_ in one _K. pneumoniae_ isolate with an MIC of 64 mg/L). Besides _fosA3_ and _fosA5_, genes encoding other fosfomycin-modifying enzymes (_fosC2, fosL1-L2_) were not found in any of the Fosfomycin-resistant strains. Also, no mutations of cell-wall transport systems (_glpT_ and _uhpT_), their respective regulatory genes (_cyaA_ and _ptsI_), nor _murA_ targets were present in any of these strains.

The rates of susceptibility to the proposed new regimens were assessed using the novel combination breakpoint thresholds determined by HFIM assays[20–22]. For the _E. coli_ and _K. pneumoniae_ isolates for which MIC determination was performed, prediction of susceptibility to at least one antibiotic combination was 77/108 (71%) isolates for flomoxef–amikacin, 85/108 (79%) for fosfomycin-amikacin and 82/108 (76%) for flomoxef–fosfomycin compared to 34/108 (31%) for ampicillin–gentamicin and 24/108 (22%) for cefotaxime (Table 1 and Supplementary Table 1, Supplementary Data 1). The three new combinations exhibited strong activity against ESBL-producing _E. coli_ and _K. pneumoniae_ isolates (excluding those strains producing an AmpC cephalosporinase and carbapenemase in addition to ESBL) and an excellent coverage was observed with flomoxef–amikacin and flomoxef–fosfomycin combinations (n = 52/52, 100%) as well as with the fosfomycin-amikacin combination (n = 48/52, 92%).

Resistance of _S. marcescens_ and ECC isolates was predicted by the presence of resistance genes. Both microorganisms harbored species-specific genes encoding for inducible chromosomal AmpC and displayed resistance to cefotaxime, as well as to flomoxef alone. In addition, genes encoding for ESBLs were observed in 6/33 (18%) of _S. marcescens_ and in 10/27 (37%) of ECC isolates. Resistance to gentamicin was predicted in 4/33 (12%) of _S. marcescens_ and in 10/27 (37%) of ECC.

The activity of amikacin was predicted to cover 91% of _S. marcescens_ and 96% of ECC isolates, since genes coding for AMEs or for 16S rRNA methylase genes that confer resistance to this antibiotic were

not widespread. Carbapenem resistance was predicted to be low, as only one isolate per species was found to carry an NDM-like gene.

Among _A. baumannii_ isolates, 91% (73/80) harbored OXA-like carbapenemase genes (most often represented by OXA-23 and OXA-58) and 45% (36/80) also co-harbored NDM-1 always in association with OXA-23, as well as resistance genes to aminoglycosides, including the 16S rRNA methylase genes, that confers high-level resistance to all aminoglycosides for clinical use.

### Characterization of _K. pneumoniae_

_K. pneumoniae_ isolates were found in all 13 sites, and WGS analysis revealed high genetic diversity with a total of 56 different sequence types (STs) found among the 135 isolates analyzed. One hundred and thirteen isolates belonging to 43 STs were predicted to be MDR, based on the presence of acquired drug resistance genes to 3 or more classes of antimicrobial agents. The geographic distribution of the _K. pneumoniae_ MDR STs notably differed by sites or by countries (Table 2, Fig. 2). The four most frequent MDR STs (ST39, ST17, ST14, and ST73) were almost exclusively found at South African sites, while ST15 and ST147 were found in Asian sites including Vietnam and Bangladesh, respectively. Other _K. pneumoniae_ MDR STs, such as ST307 or ST11 were reported from different sites in different countries/continents showing no association with a specific region or country. As shown in Table 2, _K. pneumoniae_ belonging to different STs were often co-circulating at different sites.

CgMLST revealed that isolates of the same ST found at different sites could be delineated in different clones (Fig. 2, Supplementary Data 2). This analysis had, as expected, a much higher discriminatory power than classical multilocus sequence typing and it allowed the detection of distinct clones within a single ST, most notably for ST39, ST17, and ST14.

In total, 83% (112/135) of _K. pneumoniae_ isolates carried one or several ESBL genes encoding resistance to extended spectrum cephalosporins; this was confirmed phenotypically for 65/87 isolates (44 ESBL only and 21 in association with carbapenamase) tested phenotypically. ESBL encoding genes were widely distributed and found in isolates from 11/13 sites (Fig. 3); the $bla_{CTX-M-15}$ gene was the most prevalent ESBL, present in 95 isolates at 10 sites. Other less frequently found ESBL genes were $bla_{CTX-M-14}$ (in 7 isolates), $bla_{CTX-M-27}$ (in 2 isolates) both in Vietnam, $bla_{SHV-2}/bla_{SHV-12}$ (in 6 isolates of which 5 in S. Africa and one in Bangladesh) and $bla_{TEM-53}$ (in 2 isolates, in Italy).

On the contrary, only six isolates carried AmpC genes, $bla_{MOX-2}$ (n = 3, site GR4) and $bla_{DHA-1}$ (n = 3 one each at sites BA1, TH13, and VI16).

Forty-one of the 135 _K. pneumoniae_ isolates (30.4%) carried one or several carbapenem resistance encoding genes belonging to the four major carbapenemase family types (KPC, NDM, VIM, and OXA-48 like).

Carbapenemase-producing strains were found at 7 sites across five countries (Fig. 4). The most frequent carbapenem resistance gene by far was $bla_{NDM}$ being identified in 31/135 (23%) of _K. pneumoniae_ isolates. Among the different variants, $bla_{NDM-1}$ was the most prevalent (n = 20) but other alleles ($bla_{NDM-4}$ and $bla_{NDM-5}$) were also found. As known from the epidemiology of carbapenemase producers, specific carbapenem resistance genes are associated with the geographic areas in which the isolates were found. Despite the small number of isolates, KPC producing strains were mostly found in Brazil, those carrying OXA-48-like carbapenemase in Bangladesh (BA1) and South Africa (SA11) and those with NDM-4 and NDM-5 in South-East Asia (VI16) and in Asia (BA1), respectively (Fig. 4).

Acquired aminoglycoside resistance genes were found in 122/135 (90%) _K. pneumoniae_ isolates (Supplementary Fig. 1), usually in association with ESBL- and/or with carbapenemase-coding genes. These associations were proven as significantly relevant ($p < 0.0001$) for both combinations. Namely, the _aac(3)-II_ genes which confers resistance to gentamicin were most often present in association with $bla_{CTX-M-15}$

ESBL gene (80.4%; $n = 86/107$ vs. only 10.7%; $n = 3/28$ in ESBL-negative strains; $p < 0.0001$). On the other hand, the *aph(3')-VI* gene which is known to be associated with resistance to amikacin was rare being found in only 6 isolates at three sites, BA1, GR4 and IT8 ($n = 2$ each). Genes encoding for 16S rRNA methylases (*armA, rmtB, rmtC, rmtF*), which confer high-level resistance to all clinically used aminoglycosides, were identified in 23 isolates from 5 sites (BA1, SA10, SA11, SA12, and VI16) (Supplementary Fig. 1). The 16S rRNA methylase encoding genes were carried only by carbapenemase-producing isolates, mostly NDM-producers (61.3%; $n = 19/31$ vs. only 3,8%; $n = 4/104$ carbapenemase negative or strains carrying carbapenemase genes other than NDM; $p < 0.0001$).

## Characterization of *E. coli*

Thirteen different STs were identified in the 34 *E. coli* isolates. ST1193 ($n = 13$, 38%) was the most abundant in isolates from Vietnam (VI16, $n = 5$) and one site in South Africa (SA11, $n = 4$). The second most prevalent ST type was ST131 ($n = 5$, 15%), reported from two sites in South Africa (SA10, $n = 1$ and SA12, $n = 2$), Thailand (TH13, $n = 1$) and Vietnam (VI16, $n = 1$).

Compared to *K. pneumoniae*, *E. coli* isolates carried fewer antibiotic resistance genes. Thirty-eight percentage of the isolates (13/34) harbored an ESBL gene, mostly $bla_{CTX-M-27}$ ($n = 8$) and $bla_{CTX-M-15}$ ($n = 5$) genes. Only one isolate carried a $bla_{KPC-2}$ carbapenem resistance gene (site VI16) and one other isolate had an AmpC gene ($bla_{CMY-2}$, site TH13).

Aminoglycoside resistance genes were detected in nine *E. coli* isolates from five sites. Genes encoding AMEs modifying only gentamicin (*aac(3)-II* and *ant(2')-Ia*) were observed in 7 isolates and were found in association with ESBL genes mostly in ST131 or in ST1193. Furthermore, two *E. coli* strains from site VI16 carried the *rmtB* 16S RNA methylase gene associated with resistance to all aminoglycosides including amikacin.

## Characterization of *A. baumannii*

Among all *A. baumannii* isolates analyzed, 13 different sequence types were identified, with two of them predominant: ST1 ($n = 36$) and ST2 ($n = 24$). These STs represent the two major international pandemic lineages, GC1 and GC2, respectively. Two sites from South Africa (SA10 and SA12) accounted for 80% of the *A. baumannii* isolates. Two different clusters could be delineated within ST1 and ST2 each (Fig. 5, Supplementary Data 2). The ST2 strains were delineated in two clearly distinct clusters differing between sites SA10 and SA12. The ST1 strains also grouped in two different clusters, but they were more closely related to each other and had smaller allelic loci distances (Fig. 5). Other sporadic STs not associated with pandemic lineages were found at single sites.

*A. baumannii* isolates displayed a very extensive drug resistance profile. Carbapenemase-producing *A. baumannii* isolates were found in eight sites in five countries in Asia and Africa (Fig. 6). $Bla_{OXA23}$ was the most frequently present carbapenem resistance gene, present in 64/80 (80%) of the isolates and nearly half of the OXA-23-producers ($n = 36$; 45%) co-carried $bla_{NDM-1}$. Ten isolates (13%), seven of which from site SA12, carried a $bla_{OXA-58}$ gene (Fig. 6). All ST1 isolates from sites SA10, SA11 and SA12 in South Africa always carried the $bla_{NDM-1}$ and $bla_{OXA-23}$ genes. On the other hand, strains belonging to ST2 carried $bla_{OXA-23}$ alone and never in association with $bla_{NDM-1}$. Most of the $bla_{OXA-58}$ positive isolates ($n = 10$) were associated with ST243 and were found almost exclusively at site SA12 ($n = 7$; 70%).

More than 90% (74/80) of the *A. baumannii* isolates harbored resistance genes that confer resistance to all clinically used aminoglycosides including amikacin (Supplementary Fig. 2). The *armA* rRNA 16S methylase gene (in 53/80 strains, 62%), often in association with *aac(3)-Ia* (in 35/80 strains, 44%), was the most common aminoglycoside resistance mechanism.

**Table 2 | Distribution of the 16 most prevalent MDR STs of *K. pneumoniae* by site**

| Site** | Sequence types* | | | | | | | | | | | | | | | | Total |
|---|---|---|---|---|---|---|---|---|---|---|---|---|---|---|---|---|---|
| | ST39 | ST17 | ST14 | ST73 | ST307 | ST13 | ST530 | ST11 | ST15 | ST26 | ST37 | ST147 | ST502 | ST1552 | ST1741 | ST4291 | |
| BA1 | 1 | | | | | | | 1 | | | | 3 | | | | | 5 |
| BR2 | | | | | | | | 2 | | | | | | | | | 2 |
| BR3 | | | | | | | | | | | | | | | | | 0 |
| GR4 | | | | | | | | | | | 1 | | | | | | 1 |
| IT8 | | | | | | | | | | | | | | | | | 0 |
| KE9 | | 1 | | | | | | | | | | | | | 1 | | 2 |
| SA10 | | 3 | 3 | 6 | | | 4 | | | 3 | 1 | | | | | | 20 |
| SA11 | 12 | | | | | 4 | | | | | | | 3 | 3 | | | 22 |
| SA12 | 1 | 7 | 5 | | 2 | | | | | | | | | | | 3 | 18 |
| TH13 | | | | | 2 | | | | | | | | | | | | 2 |
| TH14 | | | | | | | | | | | | | | | | | 0 |
| UG15 | | | | | 1 | | | | | | | | | | | | 1 |
| VI16 | | | | | | | | | 3 | | 1 | | | | 2 | | 6 |
| Total | 14 | 11 | 8 | 6 | 5 | 4 | 4 | 3 | 3 | 3 | 3 | 3 | 3 | 3 | 3 | 3 | 79 |

*The table includes sequence types that were found in three or more isolates.
** *VI* Vietnam, *UG* Uganda, *TH* Thailand, *SA* South Africa, *KE* Kenya, *IT* Italy, *GR* Greece, *BR* Brazil, *BA* Bangladesh. The numbers following the country keys refer to the site number. Source data are provided as a Source Data file.

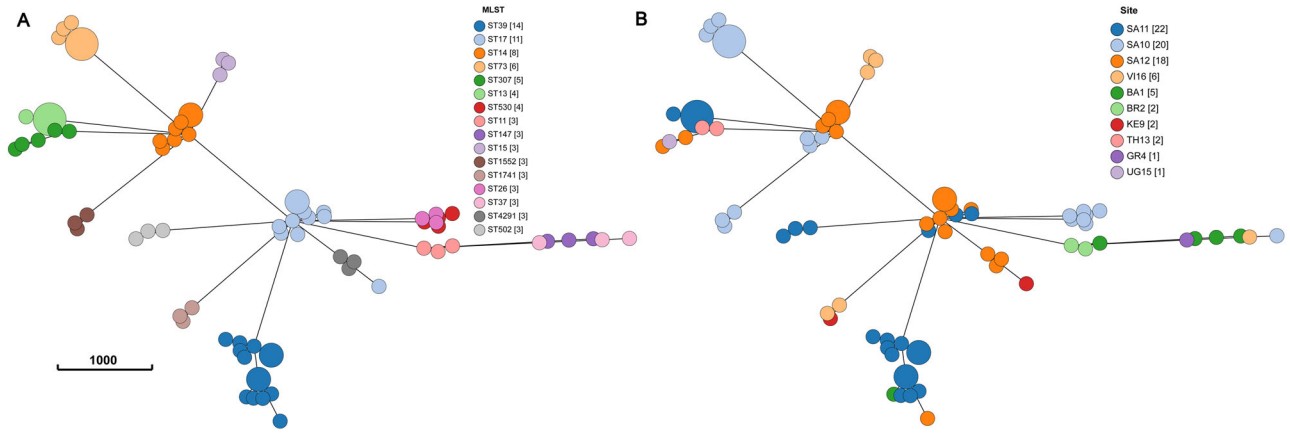

**Fig. 2 | Minimum spanning tree of cgMLST analysis of MDR clones of *K. pneumoniae* (*n* = 79) showing specific local site clustering.** STs with three or more isolates are presented. **A** ST based clustering. **B** Site based clustering. VI Vietnam, UG Uganda, TH Thailand, SA South Africa, KE Kenya, GR Greece, BR Brazil, BA Bangladesh. The numbers following the country keys refer to the site number. Source data are provided as a Source Data file.

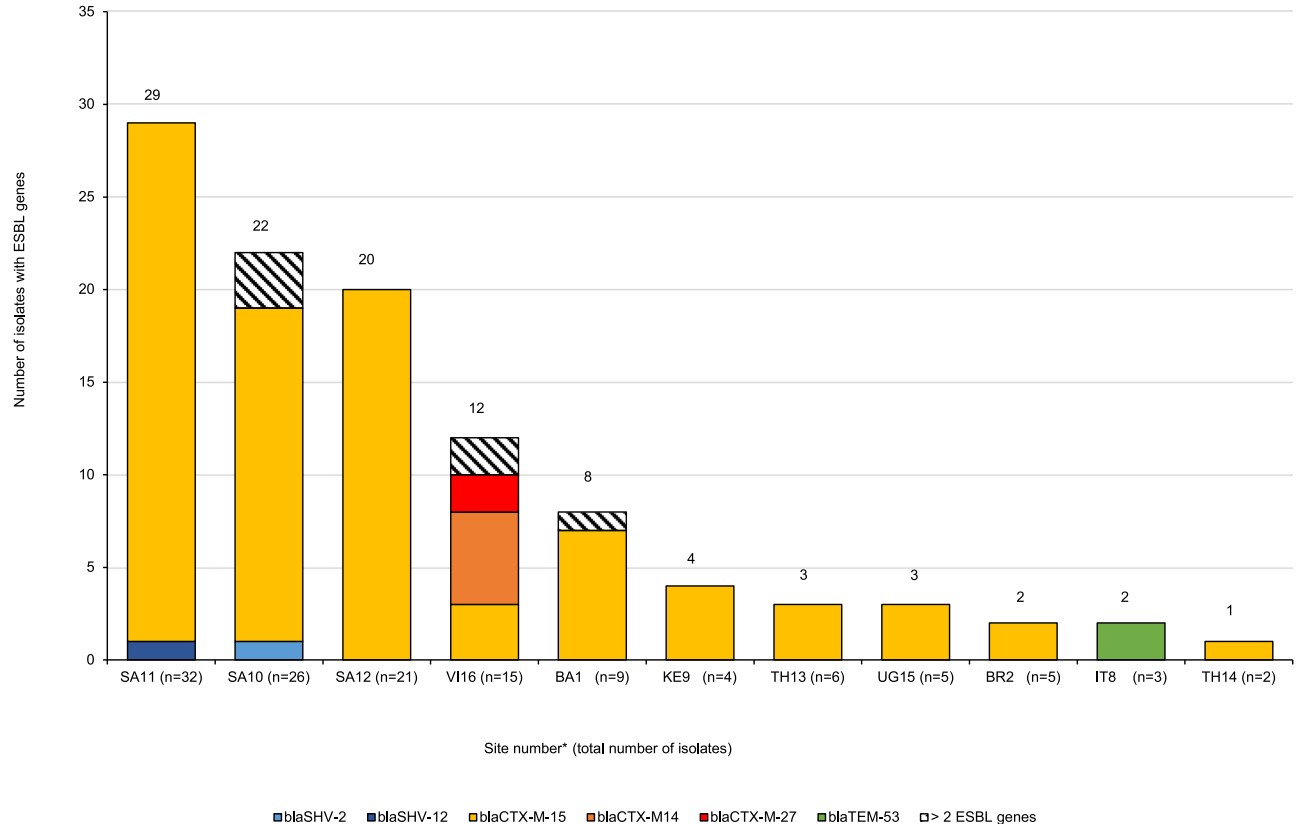

**Fig. 3 | Distribution of ESBL genes of *K. pneumoniae* isolates (*n* = 135) by site.** (*n*) = total number of *K. pneumoniae* isolates per site. *VI Vietnam, UG Uganda, TH Thailand, SA South Africa, KE Kenya, IT Italy, BR Brazil, BA Bangladesh. The numbers following the country keys refer to the site number. Source data are provided as a Source Data file.

## Characterization of ECC

ECC isolates (*n* = 27) were collected at 12 sites in 8 countries with different subspecies occurring at the different sites. *E. hormaechei* (*n* = 15) was the most common species found at 8 sites; *E. roggenkampii* (*n* = 4) at two and *E. asburiae* (*n* = 3) at 3 sites. Antimicrobial resistance in ECC is partly due to intrinsic chromosomal genes. All strains had in their core genome different alleles of *bla*ACT, an intrinsic AmpC gene that when overexpressed leads to resistance to expanded spectrum cephalosporins.

Multiple acquired resistance genes were found in 10/27 (37%) of the ECC isolates. All of these MDR strains were identified genetically as *E. hormaechei*. These strains belong to five different ST types including the well-known ST68 and ST78 MDR lineages. Nine of the 10 ESBL-producing *E. hormaechei* isolates carried a *bla*CTX-M-15 and one a *bla*SHV-12 gene. Other resistance genes that were frequently found in association with ESBLs included *bla*OXA-1 gene (resistance to piperacillin-tazobactam) and *aac(3)-II* coding genes (resistance to gentamicin). One of the 10 strains collected at site BA1 also harbored

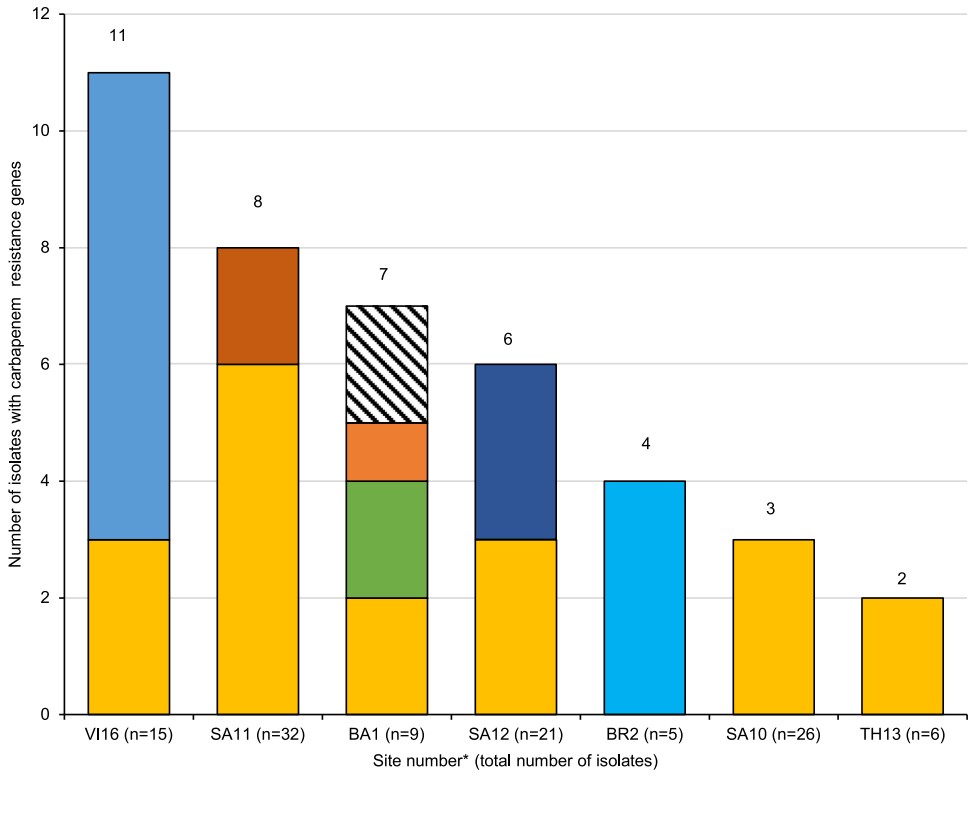

**Fig. 4 | Distribution of carbapenem resistance genes of *K. pneumoniae* isolates (*n* = 135) by site.** (*n*) = total number of *K. pneumoniae* isolates per site. Sites that collected *K. pneumoniae* isolates lacking any carbapenem resistance genes are not shown in this figure (UG15 *n* = 5, GR4 *n* = 4, KE9 *n* = 4, IT8 *n* = 3, BR3 *n* = 3 and TH14 *n* = 2). *VI Vietnam, UG Uganda, TH Thailand, SA South Africa, KE Kenya, IT Italy, GR Greece, BR Brazil, BA Bangladesh. The numbers following the country keys refer to the site number. Source data are provided as a Source Data file.

a *bla*$_{NDM-1}$ gene and a *rmtB* 16S rRNA methylase gene rendering it almost pan-resistant.

### Characterization of *Serratia marcescens*

Thirty-three *Serratia marcescens* isolates were collected from 7 sites in 6 countries, majority (*n* = 19; 58%) from the two South African sites. The cgMLST analysis highlighted the diversity of the *S. marcescens* isolates, between the different sites but also within single sites.

Genes encoding for ESBLs, *bla*$_{VEB-5}$ (*n* = 2), *bla*$_{CTX-M-14}$ (*n* = 2), and *bla*$_{CTX-M-15}$ (*n* = 2) were found in 6 (18%) isolates in Vietnam (*n* = 3), Bangladesh (*n* = 2), and Thailand (*n* = 1). One isolate from Vietnam harbored a *bla*$_{NDM-5}$ carbapenem resistance gene.

16S rRNA methylase genes (*armA* (*n* = 2); *rmtB* (*n* = 1)) were found in three isolates at two sites, BA1 and VI16.

### Discussion

The NeoOBS clinical study documented Gram-negative bacteria as the most common causative pathogens in neonatal sepsis[6], which is consistent with recent literature in LMICs[23]. The two most frequently encountered bacterial species were *K. pneumoniae* and *A. baumannii*, a finding which is also in line with data from several single and multicenter studies of neonatal sepsis in LMICs[4,5,7,9,24–26]. Among the Enterobacterales species isolates, resistance to β-lactams was due to the widespread distribution of ESBLs, especially among *K. pneumoniae* (79%) and *E. coli* (38%) isolates. Piperacillin–tazobactam, that provides partial ESBL/pseudomonal coverage and was commonly used as empirical treatment by some hospitals[6], had moderate antibacterial activity against *E. coli* (susceptibility of 67%) but low activity against *K. pneumoniae* (susceptibility of 35% only), due to the widespread

distribution of *bla*$_{OXA-1}$. These high rates of antimicrobial resistance have been translated to the increased use of meropenem for the treatment of patients with sepsis at these sites[6]. However, in this study resistance to meropenem was also observed in approximately 30% of *K. pneumoniae* isolates.

The three generic antibiotics, flomoxef, amikacin and fosfomycin, that were evaluated in combinations exhibited very good activity against *Klebsiella pneumoniae* and *E. coli* which account as two of the major pathogens in neonatal sepsis. Overall, flomoxef showed a good stability to ESBLs such as the CTX-M enzymes which were predominant in these species at all sites and was as active as the carbapenems against these organisms. Amikacin, thanks to its 1-amino-alpha-hydroxybutyrate side chain group had an improved stability to modification by most types of AMEs and consequently displayed substantially higher activity and lower rates of resistance compared to gentamicin. Of all antimicrobial agents tested, fosfomycin had the lowest rate of resistance. Fosfomycin retained excellent antibacterial activity against ESBL-producing isolates but also against a significant proportion of carbapenem-resistant *K. pneumoniae* isolates without the cross-resistance seen with the other classes of antimicrobial agents due to its unique mechanism of action[19]. Globally, all three novel antibiotic combinations of these antibiotics appeared superior in susceptibility coverage rates (i.e., at least one agent of the combination being active) to the current standard-of-care regimens advocated by the WHO, making these as suitable alternatives for the empirical treatment of neonatal sepsis in settings with high prevalence of antimicrobial resistance. The fact that these three new antibiotic combinations display comparable activity to meropenem against multidrug resistant Gram-negative isolates also highlight their interest as

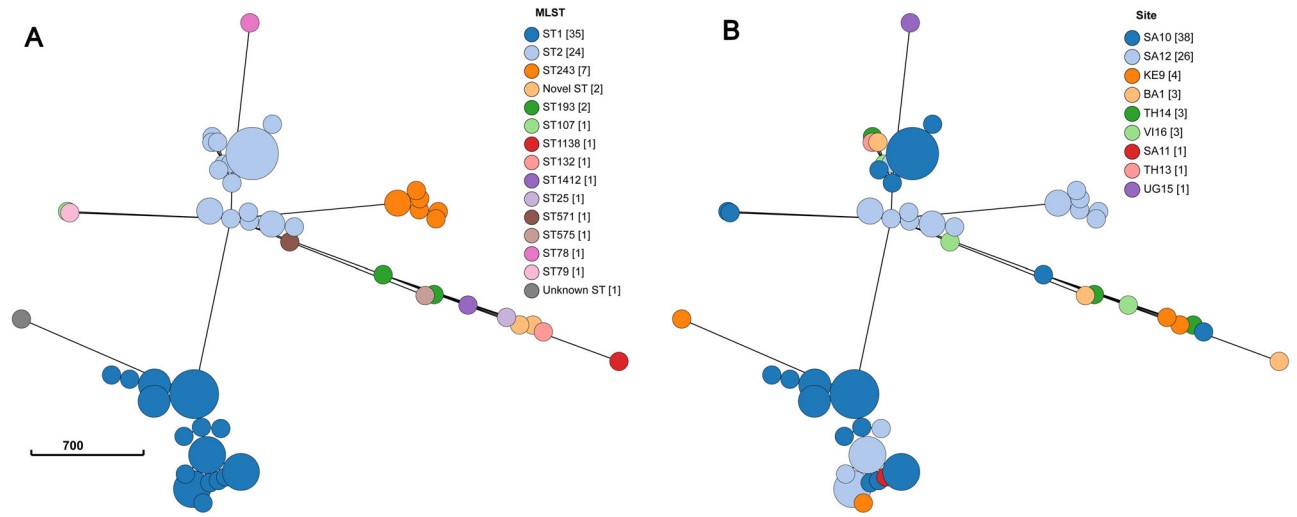

**Fig. 5 | Minimum spanning tree from *A. baumannii* genomes (*n* = 80) by cgMLST sequence types (STs). A** ST based clustering. **B** Site based clustering. VI Vietnam, UG Uganda, TH Thailand, SA South Africa, KE Kenya, BA Bangladesh. The numbers following the country keys refer to the site number. Source data are provided as a Source Data file.

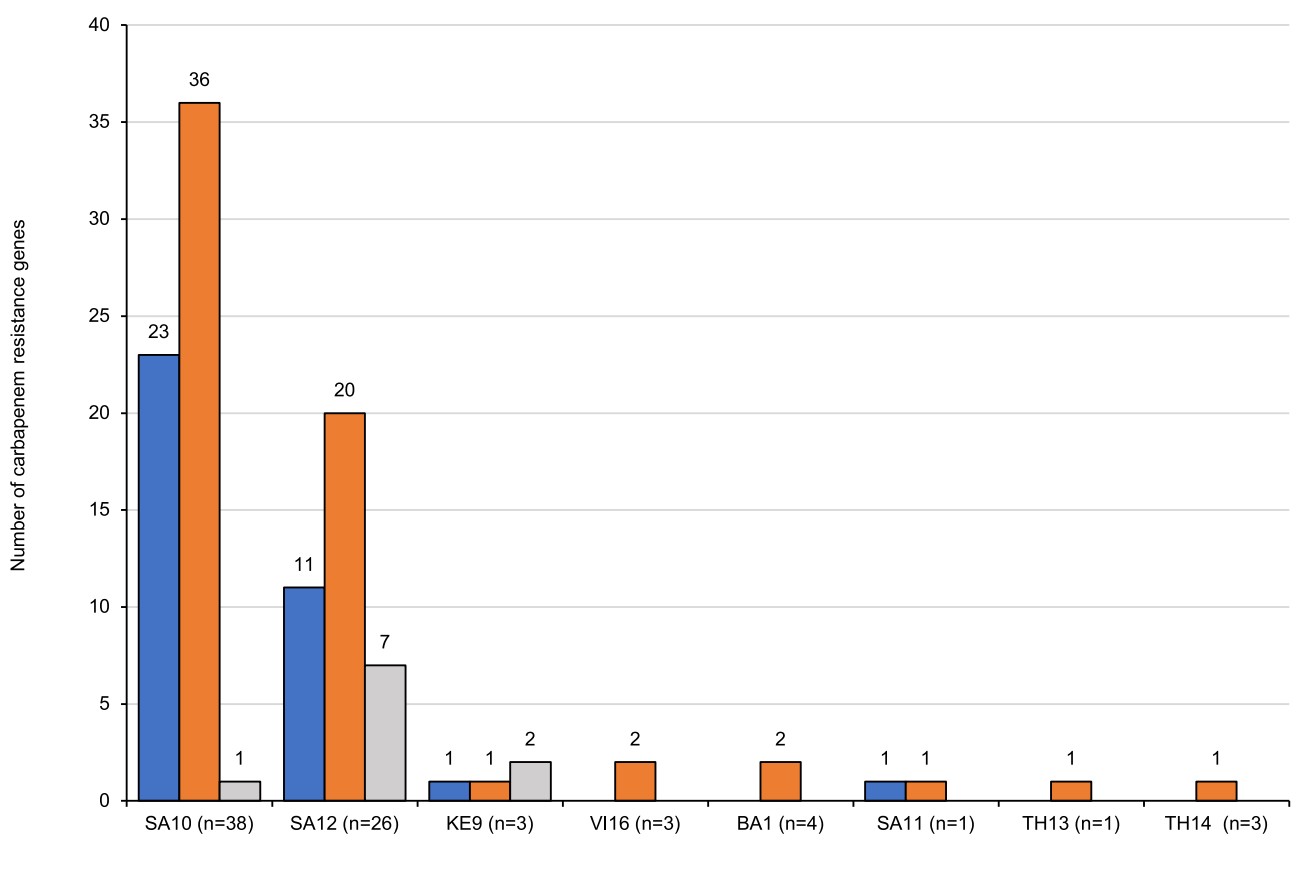

**Fig. 6 | Distribution of carbapenem resistance genes in *A. baumannii* isolates (*n* = 80) by site.** Site UG15 (*n* = 1) not represented in the chart since one unique *A. baumannii* isolate lacking any acquired carbapenemase producing genes. *VI Vietnam, UG Uganda, TH Thailand, SA South Africa, KE Kenya, IT Italy, GR Greece, BR Brazil, BA Bangladesh. The numbers following the country keys refer to the site number. Source data are provided as a Source Data file.

**Table 3 | List of sites whose isolates were characterized at University of Antwerp (UA)**

| Country | UA site number | Country/site number codes* |
|---|---|---|
| Bangladesh | Site 1 | BA1 |
| Brazil | Site 2 | BR2 |
| Brazil | Site 3 | BR3 |
| Greece | Site 4 | GR4 |
| Italy | Site 8 | IT8 |
| Kenya | Site 9 | KE9 |
| South Africa | Site 10 | SA10 |
| South Africa | Site 11 | SA11 |
| South Africa | Site 12 | SA12 |
| Thailand | Site 13 | TH13 |
| Thailand | Site 14 | TH14 |
| Uganda | Site 15 | UG15 |
| Vietnam | Site 16 | VI16 |

*The country/site number codes are used as key site codes throughout the paper.

carbapenem-sparing regimens sepsis which is very important because of the high carbapenem resistance rates that was encountered at many of the participating sites in the NeoOBS study.

All the Gram-negative isolates analyzed in this study showed wide genetic diversity, especially *K. pneumoniae* isolates, with 56 different STs identified in 135 isolates. The MDR *K. pneumoniae* isolates belonged to various international lineages and clones, that have been already widely reported both in adults and neonates, and are known to be associated with nosocomial outbreaks and endemic in hospital settings[27–29]. Besides their ability to spread through clonal expansion, these MDR lineages are also known to carry several plasmids, transposons and other mobile genetic elements that allow them to acquire and spread antimicrobial resistance within and across different bacterial species[30].

In *E. coli*, ST131 and ST1193 were the most frequent STs in the NeoOBS study. This is in line with a recent epidemiological study which found that ST131 and ST1193 are the most widely globally distributed MDR clones recorded to date[31]. Interestingly, we found that *E. coli* ST1193 was also largely distributed in different sites from Vietnam to Kenya, Uganda and South Africa, highlighting the expansion of this lineage including the African continent.

Multidrug-resistant *A. baumannii* strains were unevenly distributed at eight sites, but found in large numbers at only two of the participating sites (SA10 and SA12).

*A. baumannii* has emerged during the last decade as a major difficult-to-treat nosocomial bacterial pathogen because of its frequent multidrug resistance, its ability to persist for long time periods in the hospital environment and to colonize susceptible individuals, especially in critical care settings[32]. Unlike *K. pneumoniae* and *E. coli* which showed high genetic diversity, *A. baumannii* isolates were less diverse and mostly belonged to the two dominant pandemic international clonal groups (CG1 and CG2) and were found to occur in a limited number of sites, essentially in South Africa.

Resistance of *S. marcescens* and ECC was inferred from genotyping of resistance genes. Both species had inducible chromosomal AmpC genes and showed resistance to the WHO second-line cephalosporin regimen (cefotaxime/ceftriaxone) as well as to flomoxef alone. It is well known that derepressed mutants can be selected in AmpC inducible species during therapy with these agents and lead to the development of resistance and clinical failure, especially in invasive infection caused by ECC[33].

Although the study recruited over 3200 neonates, the limitations of this study primarily relate to the relatively small number of samples obtained, especially from sites where lower number of patients were enrolled. Besides differences in the prevalence and distribution of neonatal sepsis-associated pathogens, variation in blood culture positivity rates between centers was also observed[6], possibly indicating differences in the collection and/or performance of microbiological methods used locally. The majority of neonatal units participating in this study were based at tertiary hospitals in urban areas, so the burden of AMR may not be representative for district hospitals. These elements represent an important bias in most AMR studies in low-resource settings, where the need for high-quality microbiology means that certain settings may be overrepresented. Another limitation concerns the arbitrary breakpoints of flomoxef that we used in this study to assess the efficacy of the combinations because there is currently no formal clinical breakpoint for flomoxef approved by EUCAST or by CLSI. Despite the promising potential of the three combinations evaluated for empiric treatment of neonatal sepsis in areas with high antimicrobial resistance, there is a need for further studies concerning pharmacodynamic characterization, as not all thresholds for success of these combinations have been published.

## Methods

### Study setting

A prospective observational clinical study of neonatal sepsis was conducted between 2018 and 2020 at 19 hospitals across 11 predominantly LMIC countries from five WHO regions (Africa, Americas, Europe, Southeast Asia, and Western Pacific)[6]. Hospitalized infants <60 days of age with an episode of clinically suspected sepsis and signed informed consent form were eligible for enrollment.

Ethical approval was obtained from St. George's, University of London (SGUL) Research Ethics Committee and sites' local, central or national ethics committees and other relevant local bodies, where required. The study was registered with ClinicalTrials.gov (NCT03721302).

### Strain collection and characterization of isolates

The Laboratory of Medical Microbiology (LMM) at the University of Antwerp received bacterial isolates from blood and cerebrospinal fluid (CSF) samples of neonates with culture-confirmed sepsis from 13 participating sites in 9 countries (Table 3). LMM did not obtain bacterial isolates from three sites from India and three from China. All the participating sites followed a well-established microbiological protocol for collection, storage and shipment of the isolates to the central laboratory[18]. At LMM, species identification was verified using Microflex LT MALDI-TOF MS (Bruker Daltonics) and the MALDI Biotyper IVD reference library (2021).

### Selection of the isolates for inclusion in this microbiology study

Identification at species level was confirmed by WGS for 723 out of 1051 bacterial isolates.

For all analyses, the patient's first clinical isolate from blood or CSF was selected. In case of mixed infection, isolates belonging to different species were also included. To avoid analyzing replicates of the same bacterial clone, only isolates displaying different genetic profiles were selected for in vitro susceptibility testing. Isolates belonging to a given species for which less than 20 isolates were obtained were not further analyzed. A detailed flow diagram of the total number of isolates received at the laboratory of the University of Antwerp during the NeoOBS study and their subsequent selection to the final number of isolates included in this study is shown in Fig. 7. No statistical method was used to predetermine sample size. Statistical significance for association of resistance mechanisms was taken at $P \leq 0.05$.

### DNA isolation and whole genome sequencing (WGS)

Genomic DNA isolation was done using the MasterPure complete DNA and RNA purification kit (Epicenter, Madison, WI, USA). Sample and

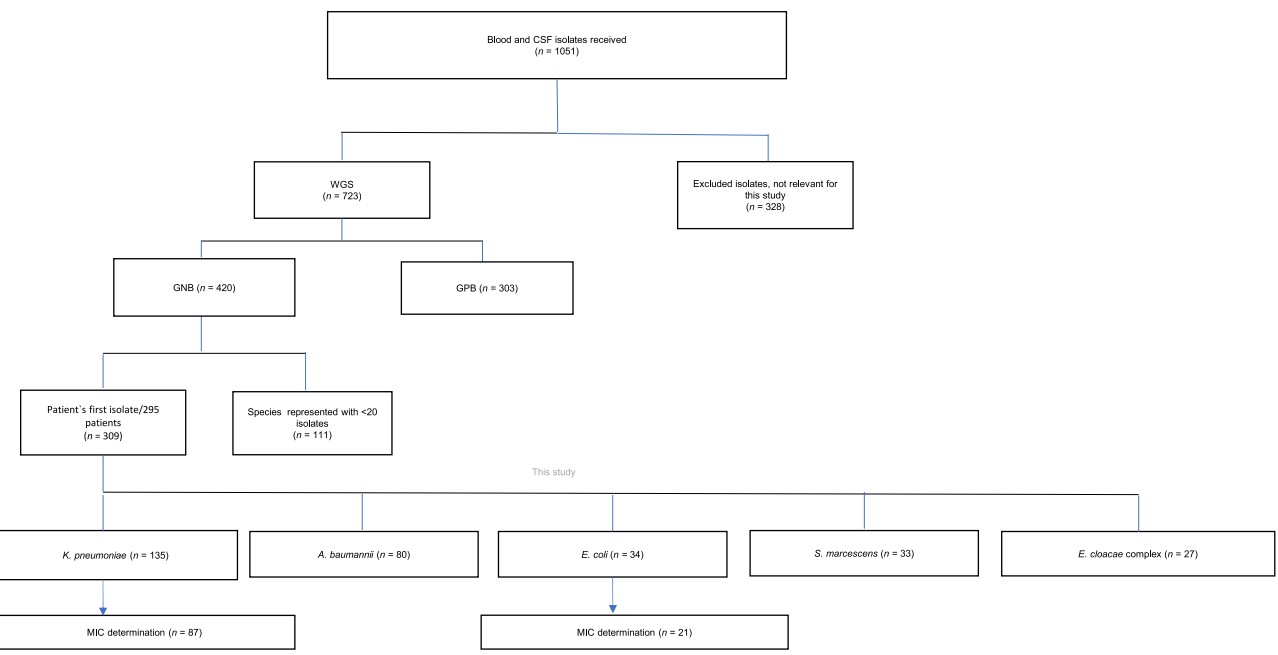

**Fig. 7 | Isolates flow diagram.** The flow diagram provides an overview of the total number of isolates received at the laboratory of the University of Antwerp during the NeoOBS study and subsequent selection to the final number of isolates included in this study. GNB gram-negative bacteria, GPB gram-positive bacteria, MIC minimum inhibitory concentration.

library preparation was done using Nextera XT sample preparation kit (Illumina). Sequencing was performed using MiSeq platform (Illumina Inc., USA), with a minimum coverage of 80-fold per strain and generated 2 × 251 bp sequence reads. Raw sequencing data were quality-assessed using FastQC and cleaned using trimmomatic v0.4.2 with default parameters for adapter removal and quality trimming. Contamination scores (lower than 5%) and the completeness (higher than 95%) were confirmed with CheckM (v1.1.6)[34]. Secondary analysis were made using an in-house developed bacterial WGS pipeline, BacPipe v.1.2.6[35] Using this pipeline, de novo assemblies of sequences were made using SPAdes (v.3.11.0) with default settings and the resulting draft genomes were annotated using Prokka (v.1.11.1). The resistomes of the strains were determined using the ResFinder v.2.1 and CARD v.5.1 databases. For MLST analysis of *E. coli*, *K. pneumoniae*, *Enterobacter* spp. and *Acinetobacter* spp., allele sequences and profile data were retrieved from PubMLST.org using the respective MLST schemes for each organism[36–39]. For core genome multilocus sequence typing (cgMLST), a gene-by-gene approach was utilized by developing a custom scheme for the specific study, assessing allelic loci distances using ChewBBACA (v.3.1.2)[40]. Clonal relatedness was defined as ≤10 allelic differences between isolates for *A. baumannii*[41] and <12 for *Klebsiella* spp.[42]. Trees were visualized using Grapetree (v.1.5.0)[43]. For strains for which phenotypic MIC susceptibility testing was not performed, the presence of one or more resistance genes to the tested antibiotics was used to reflect resistance.

### In vitro susceptibility testing
One hundred and eight isolates of *K. pneumoniae* (*n* = 87) and *E. coli* (*n* = 21) representing different genetic profiles were tested by broth microdilution method against first- and second- line WHO-recommended regimens (ampicillin, gentamicin and cefotaxime), as well as piperacillin-tazobactam, meropenem and three antibiotics (flomoxef, amikacin, and fosfomycin) that are under investigation as a potential new regimen in combination. The activity of fosfomycin was assessed by agar dilution method. All tests were performed according to the EUCAST susceptibility testing guidelines and interpretation

criteria applied in 2023 (EUCAST clinical breakpoint tables version 13.0, Jan. 1st, 2023) (https://www.eucast.org/clinical_breakpoints).

For flomoxef, in the absence of formal EUCAST/CLSI clinical breakpoint, we used an MIC susceptibility breakpoint of ≤1 mg/L, based on a large retrospective study in adult patients with bacteremia due to ESBL producers, which reported a more favorable clinical outcome in patients treated with flomoxef when isolate MICs were ≤1 mg/L compared to those with MICs ≥ 2 mg/L[44].

The susceptibility of the isolates to the proposed combination regimens was analyzed based on the novel combination breakpoint thresholds determined by the HFIM model and checkerboard assays[20–22]. If flomoxef MIC values are between 1 and 32 mg/L, combination with fosfomycin or amikacin extends flomoxef's activity, but only when the MIC of the associated drug is in the range of the corresponding MIC breakpoints, i.e., ≤32 mg/L for fosfomycin and ≤16 mg/L for amikacin[20,21]. Success of the fosfomycin-amikacin combination is predicted if the product of the two individual antibiotic MICs is ≤256 mg/L [R. da Costa, personal communication, August 31, 2023].

### Reporting summary
Further information on research design is available in the Nature Portfolio Reporting Summary linked to this article.

## Data availability
The raw sequencing data (FASTQ) generated in this study have been deposited in NCBI database under BioProject number: PRJNA1087366 and can be accessed without restrictions. See Suplementary Data 2 for exact sample, experiment and run accessions. Source data are provided in this paper.

This study also utilized publicly available databases of ResFinder v.2.1, CARD v.5.1 and PubMLST.org. Source data are provided in this paper.

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

## Acknowledgements

This study was made possible with support from Bill & Melinda Gates Foundation; German Federal Ministry of Education and Research; German Federal Ministry of Health; Government of the Principality of Monaco; the Indian Council for Medical Research; Japanese Ministry of Health, Labor and Welfare; Netherlands Ministry of Health, Welfare and Sport; South African Medical Research Council; UK Department of Health and Social Care (UK National Institute of Health Research and the Global Antimicrobial Resistance Innovation Fund—GAMRIF); and Wellcome Trust.

## Author contributions

Conceptualization: B.K.B., T.V., C.L., S.E., R.M.A.C., A.C., N.R., J.B., A.S.W., L.J.V.P., A.D., J.A.B., A.W., P.H., S.M.K., M.S., Y.G., H.G. Data curation: B.K.B., T.V., C.L., Y.G., B.B.X., R.M.A.C. Formal analysis: B.K.B., T.V., C.L., Y.G., B.B.X., R.M.A.C., T.K., H.G., L.J.V.P. Funding acquisition: S.E., S.O.'B., J.B., P.H., M.S. Investigation: B.K.B., T.V., C.L., S.E., B.B.X., T.K., R.M.A.C., S.O.'B., A.C., N.R., J.B., A.S.W., L.J.V.P., A.D., J.A.B., A.W., P.H., S.M.K., M.S., Y.G., H.G., A.R., Wo.S., J.W., H.T.B.N., R.M., C.Z., T.N., A.W., C.B.S., P.J., Wi.S., M.S.I. Methodology: B.K.B., T.V., C.L., S.E., B.B.X., T.K., R.M.A.C., S.O.'B., L.J.V.P., P.H., S.M.K., M.S., Y.G., H.G. Project administration: B.K.B., T.V., C.L., S.E., B.B.X., T.K., A.C., N.R., J.B., A.S.W., R.M.A.C., S.O.'B., A.R., Wo.S., L.J.V.P., P.H., S.M.K., M.S., Y.G., H.G., E.N.B., E.R., M.D.L., L.R., D.B., A.D., S.L., S.B., F.N., T.M.D., T.R.C., K.P., J.A.B., S.K.S., M.M.M.P., C.G.C. Software: B.K.B., T.V., B.B.X., Q.L., C.L., T.K., Y.G. Supervision: B.K.B., T.V., B.B.X., C.L., T.K., Y.G. Validation: B.K.B., T.V., B.B.X., C.L., T.K., Y.G., S.E., R.M.A.C., S.O.'B., A.C., N.R., J.B., A.S.W., A.R., Wo.S. Visualization: B.K.B., T.V., B.B.X., Q.L., C.L., T.K., Y.G. Writing—original draft preparation: B.K.B., T.V., B.B.X., C.L., Y.G., H.G. Writing—review & editing: B.K.B., T.V., B.B.X., C.L., T.K., S.E., S.O.'B., R.M.A.C., A.C., N.R., J.B., A.R., Wo.S., A.S.W., E.N.B., E.R., M.D.L., L.R., D.B., A.D., J.W., S.L., S.B., F.N., H.T.B.N., T.M.D., T.R.C., K.P., J.A.B., R.M., C.Z., T.N., A.W., C.B.S., P.J., Wi.S., S.K.S., M.M.M.P., C.G.C., M.S.I., L.J.V.P., P.H., S.M.K., M.S., Y.G., H.G.

## Competing interests

The authors declare the following competing interests: J.A.B.: Research grant support to the university from Wellcome, NIHR, MRC, and the Bill & Melinda Gates Foundation who had no role in any aspect of the study or decision to publish. A.S.W. is an NIHR Senior Investigator supported by the NIHR Biomedical Research Center Oxford and core support to the MRC Clinical Trials Unit [MC_UU_00004/05]. J.B. is an NIHR Advanced Fellow and Chief Investigator supported by grant NIHR302554 and H2020 Agreement number 965328. The remaining authors declare no competing interests.

## Additional information

Biljana Kakaraskoska Boceska ⓘ[1] ✉, Tuba Vilken[1], Basil Britto Xavier ⓘ[1,2], Tomislav Kostyanev[1,3], Qiang Lin[1], Christine Lammens[1], Sally Ellis[4], Seamus O'Brien[4], Renata Maria Augusto da Costa[4], Aislinn Cook[5], Neal Russell[5], Julia Bielicki[5,6], Amy Riddell[5], Wolfgang Stohr[7], Ann Sarah Walker[7], Eitan Naaman Berezin[8], Emmanuel Roilides ⓘ[9], Maia De Luca ⓘ[10], Lorenza Romani[10], Daynia Ballot[11], Angela Dramowski[12], Jeannette Wadula ⓘ[13], Sorasak Lochindarat[14], Suppawat Boonkasidecha[14], Flavia Namiiro[15], Hoang Thi Bich Ngoc[16], Minh Dien Tran[16], Tim R. Cressey[17], Kanchana Preedisripipat[18], James A. Berkley ⓘ[19,20,21], Robert Musyimi[22], Charalampos Zarras ⓘ[23], Trusha Nana[24], Andrew Whitelaw ⓘ[25,26], Cely Barreto da Silva[27], Prenika Jaglal ⓘ[13], Willy Ssengooba[28], Samir K. Saha[29], Mohammad Shahidul Islam ⓘ[29], Marisa Marcia Mussi-Pinhata[30], Cristina Gardonyi Carvalheiro ⓘ[30], Laura J. V. Piddock ⓘ[4], Paul T. Heath[5], Surbhi Malhotra-Kumar ⓘ[1], Michael Sharland[5], Youri Glupczynski[1] ✉ & Herman Goossens[1]

[1]Laboratory of Medical Microbiology, Vaccine & Infectious Disease Institute, University of Antwerp, Antwerp, Belgium. [2]Department of Medical Microbiology and Infection Control, University of Groningen, University Medical Center Groningen, Groningen, The Netherlands. [3]Research Group for Global Capacity Building, National Food Institute, Technical University of Denmark, Kgs, Lyngby, Denmark. [4]Global Antibiotic Research and Development Partnership

(GARDP), Geneva, Switzerland. [5]Centre for Neonatal and Pediatric Infection, Institute for Infection & Immunity, St. George's University of London, London, UK. [6]Paediatric Research Centre, University of Basel Children's Hospital, Basel, Switzerland. [7]MRC Clinical Trials Unit, University College London, London, UK. [8]Pediatric Infectious Diseases Unit, Santa Casa de Sao Paulo, Sao Paulo, Brazil. [9]Infectious Diseases Unit, 3rd Dept Paediatrics, School of Medicine, Faculty of Health Sciences, Aristotle University and Hippokration General Hospital, Thessaloniki, Greece. [10]Infectious Disease Unit, Bambino Gesu Children's Hospital, Rome, Italy. [11]Department of Pediatrics and Child Health, School of Clinical Medicine, Faculty of Health Sciences, University of the Witwatersrand, Johannesburg, South Africa. [12]Department of Paediatrics and Child Health, Faculty of Medicine and Health Sciences, Stellenbosch University, Cape Town, South Africa. [13]Department of Clinical Microbiology & Infectious Diseases, National Health Laboratory Services, CH Baragwanath Academic Hospital, Faculty of Health Sciences, University of Witwatersrand, Johannesburg, South Africa. [14]Queen Sirikit National Institute of Child Health, Bangkok, Thailand. [15]Mulago Specialized Women's and Neonatal Hospital, Kampala, Uganda. [16]Vietnam National Children's Hospital, Hanoi, Vietnam. [17]AMS-PHPT Research Collaboration, Faculty of Associated Medical Sciences, Chiang Mai University, Chiang Mai, Thailand. [18]Chiangrai Prachanukroh Hospital, Chiang Rai, Thailand. [19]Clinical Research Department, KEMRI-Wellcome Trust Research Programme, Kilifi, Kenya. [20]Centre for Tropical Medicine & Global Health, Nuffield Department of Medicine, University of Oxford, Oxford, UK. [21]The Childhood Acute Illness & Nutrition (CHAIN) Network, Nairobi, Kenya. [22]Department of Microbiology, KEMRI-Wellcome Trust Research Programme, Kilifi, Kenya. [23]Microbiology Department, Hippokration General Hospital, Thessaloniki, Greece. [24]Department of Clinical Microbiology and Infectious Diseases, School of Pathology, Faculty of Health Sciences, University of the Witwatersrand, Johannesburg, South Africa. [25]Division of Medical Microbiology, Faculty of Medicine and Health Sciences, Stellenbosch University, Cape Town, South Africa. [26]Microbiology Laboratory, National Health Laboratory Service, Tygerberg Hospital, Cape Town, South Africa. [27]Infection Control and Prevention Service, Santa Casa de Sao Paulo, Sao Paulo, Brazil. [28]Makerere University, Department of Medical Microbiology, Kampala, Uganda. [29]Child Health Research Foundation (CHRF), Dhaka, Bangladesh. [30]Department of Pediatrics, Ribeirao Preto Medical School, University of Sao Paulo, Sao Paulo, Brazil. ✉e-mail: Biljana.KakaraskoskaBoceska@uantwerpen.be; gyglupczynski@gmail.com

