## [Peer Review File · Nature Communications]

REVIEWER COMMENTS

Reviewer #1 (Remarks to the Author):

Identification of potential novel combination antibiotic regimens based on drug susceptibility and genetic diversity of Gram-negative bacteria causing neonatal sepsis in low- and middle-income countries

Boceska et al

5th January 2024

General comments and recommendation:

The manuscript presents the results of a microbiology sub-study of the international NeoOBS study. The impact of antimicrobial resistance on outcomes from neonatal sepsis is of importance to the broad readership of the journal, and a multinational dataset of this kind is of substantial value. The manuscript is well written and there is a great deal of valuable data. I was a little confused by the size of the dataset analysed here which appears larger than that reported by the 'parent' NeoOBS study.

Major comments:

The title could be clearer. It doesn't appear that the novel combinations referred to in the paper were 'identified' by the susceptibility testing and genomic analysis here, rather they have previously been identified and included in the NeoSep1 trial, prompting the analysis done here. Further, their identification wouldn't have been made by the 'genetic diversity' of the bacteria, even if it followed the genomic analysis.

The study here alludes to analysis undertaken on 723 isolates (of a total 1051 isolates, line 148). The published NeoOBS study reports 564 culture positive sepsis cases, of which 355 were gram-negatives. I don't understand this discrepancy. I found it hard to follow exactly which isolates were included in the various analyses and a flow chart would make this clearer.

Some of the language could be clarified. Not all isolates underwent susceptibility testing and WGS. The authors conflate susceptibility testing using BMD/agar dilution and resistance inferred from genomics data, though their data show the imperfections of this. It would be better to more clearly distinguish these analyses. In vitro susceptibility testing only appears to have been done on 108 Enterobacterales (87 *K. pneumo* and 21 *E. coli*). The sections reporting these data also report the genomic analysis (and 'predicted' resistance) of *S. marcescens* and *A. baumannii* for example.

Methods:

I couldn't find reference to whether and where genomic data had been uploaded to a public database.

Line 188 'combination with Fosfomycin or amikacin extends flomoxef's spectrum of activity...' seems incorrect. Isn't the purpose of the combination to improve activity against a specific organism? Unless the 'spectrum' in question refers to a spectrum of antimicrobial resistance mechanisms. On this point, can the authors provide published data to support the statement "Success of fosfomycin-amikacin

combination is predicted if the product of the two individual antibiotic MICs is ≤ 256 mg/L" (line 190).

Results:

The Minimum spanning trees could be combined to show clustering of STs and sites in one panel.

Discussion:

The discussion could highlight the key findings of this sub-study more clearly at its outset. It is somewhat difficult to track the key findings down through the discussion.

It would be more accurate to refer to *K. pneumoniae* and *E. coli* rather than Enterobacterales overall in reference to the activity of the novel combinations, acknowledging that these are the predominant species.

The limitations of the breakpoints used for the combination drug testing should be acknowledged. There are no formal EUCAST/CLSI clinical breakpoint for flomoxef and not all of the thresholds derived for 'success' of the combinations are published.

Minor comments:

The full list of the ESBL genes identified would be informative in the supplement.

Figure 1 requires labels

Reviewer #2 (Remarks to the Author):

Reviewer #3 (Remarks to the Author):

This is a very interesting well written manuscript about Gram-negative bacteria causing neonatal sepsis in LMIC.

The biggest challenge when reading was assigning the numbers in the paper.

Abstract:

Line 83 ff: I guess the denominator for the following numbers is 108, but this is not given in the abstract.

Material and Methods

Line 172: Please give the sum of 87 and 21, as the 108 pathogens are a very important number in the paper.

Results

Line 219: Here you name the number of carbapenemases but you describe them in detail much later. Please refer to the back part of the article.

Line 226: Fosfomycin 90% susceptible, but in line 247ff the combination is only 79% and 76% susceptible.

Do I misunderstand something? Can you explain this?

Line 283: It would be helpful to make clear, that there are 135 isolates and 87 are phenotypically tested. Perhaps you can explain in more detail, why exactly 87 isolates are tested?

Line 320: Please repeat here how many carbapenemases there are.

Line 358: Please give here the number of isolates (34) and 21 phenotypically tested.

Figures and Tables:

Suppl. Fig 1: I would move this to the paper, as the countries are important.

Reviewer #4 (Remarks to the Author):

Dear authors

this paper addresses surveillance and analysis of isolates from neonate sepsis cases at a range of sampling regions in the world.

the paper holds interest but needs to address issues with English usage and syntax in order to refine the text for the reader.

line 106 in approximately 40% of episodes

line 163- for the databases here did you run these through the electronic pipelines or extract the sequences and interrogate them individually against the databases to determine the sequence types? please clarify or detail better here how this was done.

I am also assuming all software used in assembly used default settings please confirm.

Please indicate reads, total length and coverage for the genomes - data that is most suitable for a genome table.

Also please indicate where these genomes have been deposited and their respective bio projects and link the identifiers in a table/ supplementary table of data for the strains. Having reviewed the supplementary material I do not see any information on the strains that were sequenced for this project.

line 196 determine by the HFIM model

line 188 but only when the MIC of the

line 190 success of the Fosfomycin

line 202 and Fig 1 - if you are not going to include the key to the sites in the figure - you still need to refer to the key somewhere ...

please ensure you have spelled out the acronym before you use it such as GNB and AMEs

line 231 bla NDM-1 gene but was found to be susceptible to

line 233 all strains carrying genes encoding for enzymes modifying gentamicin and/or

line 234 all strains encoding for 16S
line 240 genes encoding other
line 254 please add "were observed" to the end of the sentence otherwise the sentence just hangs
line 270 genes encoding for inducible
line 271 in addition, genes encoding for ESBLs
line 278 most often represented by ..
line 279 so when you say here "always in association" do you mean both genes were always present?
line 280 is a little vague and needs more information

line 286 One hundred and thirteen isolates
line 290 were primarily found in Asian sites including Vietnam and ...
line 291 Other *K. pneumoniae* MDT STs such as ST 307 or ...
line 292 finish the sentence with .. in different continents showing no association (or affiliation) with a specific region or country.
line 298 again refer the key for the regions to a supplement table
can you please clarify the * comment better - I assume you mean that you had to have 3 or more isolates across the entire collection in a ST to include them in this table.

line 314 encoding genes were widely
line 316 genes included *bla_{CTXM} 14* ... in Vietnam and *bla_{SHV}* .. are the latter also from Vietnam ?
line 320 all strains harboring carbapenemase encoding genes were phenotypically resistant by broth microdilution (please clarify this)
line 325 resistance genes are associated with the
line 336 refer to the key for regions
line 338 usually in association with (is this a significant relationship?)

line 342 genes encoding for
line 355 I am not sure you can say this here since some of these drugs are still used in human health - or do you mean they are not relevant as a line of treatment for sepsis?
line 366 one harbored and AmpC gene
line 370 methylase gene associated with resistance to all aminoglycosides
line 374 among all *A. baumannii* isolates examined, 13 different sequence types
line 392 seen of which from site SA12 carried a *bla_{oxa}* ...
line 400 key for sites needs to be referenced
line 411 are grouped together. Several isolates carried more than one
line 425 with ESBLs included *bla_{oxa}*
line 427 at site BA 1 also harbored a *bla_{NDM}*
line 434 genes encoding for ESBLs
line 446 isolates species include *K. pneumoniae* ...
line 460 56 different STs identified in 135 isolates
line 462 and endemic in hospital settings
line 478 and were found to occur in a limited number of sites mostly linked with South Africa
line 486 this study primarily relate to .. where lower numbers of patients were enrolled

line 488 distribution of neonatal sepsis-associated pathogens

line 497 showed a wide variety of bacterial species as causative agents of neonatal sepsis

line 499 distributed among the sites studied (or sampled or monitored)

line 500 second line WHO recommended treatments

REVIEWER COMMENTS

Reviewer #1 (Remarks to the Author):

Identification of potential novel combination antibiotic regimens based on drug susceptibility and genetic diversity of Gram-negative bacteria causing neonatal sepsis in low- and middle-income countries

Boceska et al

5th January 2024

General comments and recommendation:

The manuscript presents the results of a microbiology sub-study of the international NeoOBS study. The impact of antimicrobial resistance on outcomes from neonatal sepsis is of importance to the broad readership of the journal, and a multinational dataset of this kind is of substantial value. The manuscript is well written and there is a great deal of valuable data. I was a little confused by the size of the dataset analysed here which appears larger than that reported by the 'parent' NeoOBS study.

Major comments:

1. The title could be clearer. It doesn't appear that the novel combinations referred to in the paper were 'identified' by the susceptibility testing and genomic analysis here, rather they have previously been identified and included in the NeoSep1 trial, prompting the analysis done here. Further, their identification wouldn't have been made by the 'genetic diversity' of the bacteria, even if it followed the genomic analysis.

Author: This point is well taken. In accordance with this reviewer's recommendation, we have modified the title which we think better reflects the content of our paper: "Assessment of three potential combinations of antibiotic regimens against genetically diverse Gram-negative bacteria causing neonatal sepsis in low- and middle-income countries" (Line 4-6 in the revised manuscript).

2. The study here alludes to analysis undertaken on 723 isolates (of a total 1051 isolates, line 148). The published NeoOBS study reports 564 culture positive sepsis cases, of which 355 were gram-negatives. I don't understand this discrepancy. I found it hard to follow exactly which isolates were included in the various analyses and a flow chart would make this clearer.

Author: There are several reasons that might account for differences and discrepancies between the number of culture positive sepsis cases (N. Russel et al. Plos Medicine 2023; doi: 10.1371/journal.pmed.1004179) with those reported in the present study. Several of the challenges referring to the microbiology processes have been addressed separately in another article (A. Riddell et al. Antibiotics 2023; doi: 10.3390/antibiotics12050923). Local sites were requested to store all isolates from blood and CSF that were collected during the study period and the number of isolates collected by each site differed considerably due to variations in the number of patients enrolled and routine practices in repeat cultures being taken and culture-positivity rates. Identification and storage of study participant isolates required close communication between clinical study teams and laboratory teams who needed to know which patients were enrolled in the study and which isolates had to be stored. This proved to be a very challenging task. In the published clinical study, only clinically relevant isolates were taken in consideration. Since our paper deals with the microbiological characteristics, we had to manage the above mentioned limitations and started our "story" with the total number of received isolates (n=1051), afterwards explaining in the material

and methods step by step how we ended up with the number of isolates analyzed with different analyses (genomic and phenotypic analysis). For the sake of clarity, please see the flow chart which we integrated as Fig. 1 in the main manuscript (Line 168 in the revised manuscript).

3. Some of the language could be clarified. Not all isolates underwent susceptibility testing and WGS. The authors conflate susceptibility testing using BMD/agar dilution and resistance inferred from genomics data, though their data show the imperfections of this. It would be better to more clearly distinguish these analyses. In vitro susceptibility testing only appears to have been done on 108 Enterobacterales (87 *K. pneumo* and 21 *E. coli*). The sections reporting these data also report the genomic analysis (and 'predicted' resistance) of *S. marcescens* and *A. baumannii* for example.

Author: Susceptibility testing using BMD/agar dilution methods was performed on a subset of selected isolates. The selection was performed on two levels: first, on the species level where we considered, also budget wise, species that can be targeted for therapy with the investigated antibiotic combinations; second criteria was to avoid analyzing replicates of the same bacterial clone at local site, so only isolates displaying different genetic profiles were selected for in vitro susceptibility testing.

Methods:

4. I couldn't find reference to whether and where genomic data had been uploaded to a public database.

Author: The assembled genome data are deposited at NCBI with BioProject ID: PRJNA1087366. Section "Data Availability" is added in the main manuscript (Line 626-627 in the revised manuscript).

5. Line 188 'combination with Fosfomycin or amikacin extends flomoxef's spectrum of activity...' seems incorrect. Isn't the purpose of the combination to improve activity against a specific organism? Unless the 'spectrum' in question refers to a spectrum of antimicrobial resistance mechanisms.

Author: We have made the suggested change: "...combination with fosfomycin or amikacin extends flomoxef's activity, but only..." (Line 221 in the revised manuscript).

6. On this point, can the authors provide published data to support the statement "Success of fosfomycin-amikacin combination is predicted if the product of the two individual antibiotic MICs is ≤ 256 mg/L" (line 190).

Author: These are novel data and work is still not published, additional experiments need to be done. The communicated author will provided a permission for the data to be published as well as a short summary/explanation on the available results. Below please find the added explanation:

"Our group first assessed the utility of fosfomycin and amikacin as a potential alternative regimen for treating neonatal sepsis. The results were published in Darlow et al., 2021. Amikacin combined with fosfomycin for treatment of neonatal sepsis in the setting of highly prevalent antimicrobial resistance. Antimicrob Agents Chemother 65: e00293-21 (<https://doi.org/10.1128/AAC.00293-21>). This antibiotic combination was tested at clinically relevant dosage against nine Enterobacterales strains with genotypically well-characterized resistance mechanisms by hollow fiber infection model (HFIM) experiments. The obtained data promoted the proposal of a novel combination breakpoint threshold for the fosfomycin-amikacin combination against Enterobacterales: $MIC_F \times MIC_A \leq 256$ (where MIC_F and MIC_A are the fosfomycin and amikacin MICs, respectively).

To assure that the proposed new breakpoint threshold succeeds against clinical isolates from the target population to be treated with this combination, we extended the HFIM experiments against two isolates that were collected during the NeoObs study. The summary of fosfomycin and amikacin MICs for both isolates and the success and failure of the combination are shown in the table 1. The new data confirms the proposed combination breakpoint threshold, $MIC_F \times MIC_A \leq 256$. Our group is working on the draft of a manuscript to make this new data publicly available.

Table 1- Summary of the fosfomycin and amikacin MICs for the two clinical isolates collected during the NeoObs study and the HFIM outcome.

Bacteria	Strain number	MIC (mg/L)			HFIM - Success
		AMK	FOS	AMK × FOS	
K. pneumoniae	NeoObs001	8	32	256	Yes
	NeoObs002	8	64	512	No

*AMK= amikacin; FOS= fosfomycin”

Results:

7. The Minimum spanning trees could be combined to show clustering of STs and sites in one panel.

Author: Taking the reviewer’s suggestion into consideration, we tried pipelines of PHYLOViZ as well as GrapeTree to re-visualize the MLST tree. However, in either of the two pipelines, there are no options to combine both information of the site and the sequencing type into one tree to visualize. Now, we put the two plots side by side in a single figure panel to improve the readability (Lines 351 and 457 in the revised manuscript).

Discussion:

8. The discussion could highlight the key findings of this sub-study more clearly at its outset. It is somewhat difficult to track the key findings down through the discussion.

Author: This point is well taken. In line with the recommendation of this reviewer, discussion has been restructured so as to make the key findings more apparent (Lines 547-566 in the revised manuscript).

9. It would be more accurate to refer to *K. pneumoniae* and *E. coli* rather than Enterobacterales overall in reference to the activity of the novel combinations, acknowledging that these are the predominant species.

Author: Changes have been made in Table 2 (Line 300-303 in the revised manuscript), also throughout the text and in the ‘Discussion’ section.

10. The limitations of the breakpoints used for the combination drug testing should be acknowledged. There are no formal EUCAST/CLSI clinical breakpoint for flomoxef and not all of the thresholds derived for ‘success’ of the combinations are published.

Author: This point is well taken. Another limitation is added in the ‘Discussion’ section (Lines 607-613).

Minor comments:

11. The full list of the ESBL genes identified would be informative in the supplement.

Author: We have provided a new figure that clarifies the different ESBL genes identified and their distribution by sites (Fig. 4, Line 370 in the revised manuscript).

12. Figure 1 requires labels

Author: We have made the suggested change (in the revised manuscript is Fig 2, Line 235).

Reviewer #2 (Remarks to the Author):

Reviewer #3 (Remarks to the Author):

This is a very interesting well written manuscript about Gram-negative bacteria causing neonatal sepsis in LMIC.

The biggest challenge when reading was assigning the numbers in the paper.

Author: A similar remark was made by Reviewer 1 (Comment #2). A flow chart has been added in the main text to clarify and facilitate the understanding of the number of isolates characterized and analyzed in the study (Line 169 in the revised manuscript).

Abstract:

1. Line 83 ff: I guess the denominator for the following numbers is 108, but this is not given in the abstract.

Author: We have added the denominators, 87 for *Klebsiella* and 21 for *E. coli*. (Lines 86 and 89 of the revised version of the manuscript.)

Material and Methods

2. Line 172: Please give the sum of 87 and 21, as the 108 pathogens are a very important number in the paper.

Author: We have made the suggested change (Line 204 of the revised version of the manuscript).

Results

3. Line 219: Here you name the number of carbapenemases but you describe them in detail much later. Please refer to the back part of the article.

Author: This point is well taken. We have modified the text, stating here that carbapenem resistance was associated with the presence of carbapenemase encoding genes in all strains. Detailed characterization of the various carbapenemase genes is indeed reported below in the 'Characterization of *K. pneumoniae*' section (Lines 379-390).

4. Line 226: Fosfomycin 90% susceptible, but in line 247ff the combination is only 79% and 76% susceptible. Do I misunderstand something? Can you explain this?

Author: The in vitro susceptibility of fosfomycin is based on EUCAST breakpoints determined for single drug. However, neither fosfomycin nor any other antibiotic tested is aimed for use as single agent in the treatment of neonatal sepsis. Thus, one cannot compare on the one hand these single-agent susceptibility/resistance breakpoints with the percentage coverage provided for each of the combinations (combination coverage meaning that at least one of the two drugs must be active) and where each combination uses specific cutoff thresholds based on treatment success and failure in HFIM model experiments.

5. Line 283: It would be helpful to make clear, that there are 135 isolates and 87 are phenotypical tested. Perhaps you can explain in more detail, why exactly 87 isolates are tested?

Author: The total number '135' is mentioned in the sentence (Line 326 of the revised version of the manuscript). Phenotypical analysis are not mentioned in "Characterization of *K. pneumoniae*" section because this section is about the genomic characterization and does not report any phenotypical analysis.

Regarding the second part of the question, the explanation is given in the "Material and Methods" section (Lines 162-165 of the revised version of the manuscript).

Reviewer 1 made a similar remark suggesting to more clearly distinguish these analyses. Please refer to our reply to Reviewer #1, Comment #3.

6. Line 320: Please repeat here how many carbapenemases there are.

Author: This remark is well taken and an explanation is added in the text: "Forty-one of the 135 *K. pneumoniae* isolates (30.4%) carried one or several carbapenem resistance encoding genes belonging to the four major carbapenemase family types (KPC, NDM, VIM and OXA-48 like) (Lines 379-381 of the revised version of the manuscript).

7. Line 358: Please give here the number of isolates (34) and 21 phenotypically tested.

Author: We have added in the text the number of the total *E. coli* isolates analysed by WGS (n=34), but we did not mention the number of isolates subjected to phenotypical analysis because this section is about the genomic characterization of *E. coli* and does not report any phenotypical analysis (Line 433 of the revised version of the manuscript).

Figures and Tables:

8. Suppl. Fig 1: I would move this to the paper, as the countries are important.

Author: We assume you meant Suppl. Table 1, since there is no Suppl. Figure 1. As suggested by the reviewer, we moved Suppl. Table 1 into the main text as Table 1 (Line 156 of the revised version of the manuscript) and subsequent changes numbering of the following tables are made along with the corrections throughout the text.

Reviewer #4 (Remarks to the Author):

Dear authors

this paper addresses surveillance and analysis of isolates from neonate sepsis cases at a range of sampling regions in the world.

the paper holds interest but needs to address issues with English usage and syntax in order to refine the text for the reader.

1. line 106 in approximately 40% of episodes

Author: We have made the suggested change (Line 116 of the revised version of the manuscript).

2. line 163- for the databases here did you run these through the electronic pipelines or extract the sequences and interrogate them individually against the databases to determine the sequence types? please clarify or detail better here how this was done.

Author: This has been restructured and reformulated in the text (Lines 171-191 in the revised version of the manuscript).

3. I am also assuming all software used in assembly used default settings please confirm.

Author: This has been restructured and reformulated in the text (Line 182 in the revised version of the manuscript).

4. Please indicate reads, total length and coverage for the genomes - data that is most suitable for a genome table.

Author: 4. Each strain was sequenced with a minimum coverage of 80-fold, producing two sets of 251 bp sequence reads. We have added the suggested explanation in the text: "...with a minimum coverage of 80-fold per strain and generated 2x 251 bp sequence reads." (Lines 179-180 of the revised version of the manuscript).

5. Also please indicate where these genomes have been deposited and their respective bio projects and link the identifiers in a table/ supplementary table of data for the strains. Having reviewed the supplementary material I do not see any information on the strains that were sequenced for this project.

Author: The assembled genome data are deposited at NCBI with BioProject ID: PRJNA1087366. Section "Data Availability" is added in the main manuscript (Line 626-627 of the revised version of the manuscript).

6. line 196 determine by the HFIM model

Author: We have made the suggested change (Line 219 of the revised version of the manuscript).

7. line 188 but only when the MIC of the

Author: We have made the suggested change (Line 221 of the revised version of the manuscript)

8. Line 190 success of the Fosfomycin

Author: We have made the suggested change (Line 223 of the revised version of the manuscript).

9. line 202 and Fig 1 - if you are not going to include the key to the sites in the figure - you still need to refer to the key somewhere ...

Author: This point is well received and key to the site and site number is added to all figures and tables whenever applicable.

10. please ensure you have spelled out the acronym before you use it such as GNB and AMEs

Author: Thank you, it is spelled out (Line 113 for GNB and Lines 120-121 for AMEs of the revised version of the manuscript)

11. line 231 bla NDM-1 gene but was found to be susceptible to

Author: We have made the suggested change (Line 274 of the revised version of the manuscript)

12. - line 233 all strains carrying genes encoding for enzymes modifying gentamicin and/or

- line 234 all strains encoding for 16S

- line 240 genes encoding other

- line 254 please add "were observed" to the end of the sentence otherwise the sentence just hangs

- line 270 genes encoding for inducible

- line 271 in addition, genes encoding for ESBLs

- line 278 most often represented by ..

Author: All above mentioned editorial changes (syntax/orthographic) have been made throughout the text.

13. line 279 so when you say here "always in association" do you mean both genes were always present?

Author: Yes, that is what we meant.

14. line 280 is a little vague and needs more information

Author: We have changed the statement "... several other classes of antimicrobials" with "...aminoglycosides" (Line 321 of the revised version of the manuscript).

15. line 286 One hundred and thirteen isolates

Author: We have made the suggested change: 'and' is added to 'One hundred and thirteen isolates' (Line 327 of the revised version of the manuscript).

16. line 290 were primarily found in Asian sites including Vietnam and ...

Author: We have made the suggested change: (Line 332 of the revised version of the manuscript).

17. line 291 Other K. pneumoniae MDT STs such as ST 307 or ...

Author: We have made the suggested change (Line 333 of the revised version of the manuscript).

18. line 292 finish the sentence with .. in different continents showing no association (or affiliation) with a specific region or country.

Author: We have implemented the suggested addition (Line 334 of the revised version of the manuscript).

19. line 298 again refer the key for the regions to a supplement table

Author: This point is well received and key to the site and site number is added to all figures and tables whenever applicable.

20. can you please clarify the * comment better - I assume you mean that you had to have 3 or more isolates across the entire collection in a ST to include them in this table.

Author: Yes, that is what we meant. We have rephrased the sentence in the manuscript: "The table includes sequence types that were found in three or more isolates" (Line 340 of the revised version of the manuscript).

21. line 314 encoding genes were widely

Author: We have made the suggested change (Line 365 of the revised version of the manuscript).

22. line 316 genes included bla_{CTXM14} ... in Vietnam and bla_{SHV} .. are the latter also from Vietnam ?

Author: We have made the clarification in the text (Lines 367-369 of the revised version of the manuscript).

We have also added a Figure for ESBL genes (Fig.4, Line 370 of the revised version of the manuscript). The figure shows the number and distribution of ESBL genes across the different sites.

23. line 320 all strains harboring carbapenemase encoding genes were phenotypically resistant by broth microdilution (please clarify this)

Author: We have restructured the paragraph (Lines 379-381 of the revised version of the manuscript).

24. line 325 resistance genes are associated with the

Author: We have made the suggested change (Line 386 of the revised version of the manuscript).

25. line 336 refer to the key for regions

Author: This point is well received and key to the site and site number is added to all figures and tables whenever applicable.

26. line 338 usually in association with (is this a significant relationship?)

Author: The explanation for the association significance is added in the text (Lines 404-408 and Lines 414-416 of the revised version of the manuscript).

27. line 355 I am not sure you can say this here since some of these drugs are still used in human health - or do you mean they are not relevant as a line of treatment for sepsis?

Author: Indeed, this refers to the fact that these particular aminoglycosides agents are not used anymore for the treatment of any systemic infections in humans including neonatal sepsis. We have made the clarification (Lines 424-425 of the revised version of the manuscript).

Author: We agree with all below listed changes and the same have been made throughout the text:

28. line 342 genes encoding for

Author: Line 414 of the revised version of the manuscript.

29. line 366 one harbored and AmpC gene

Author: Line 440 of the revised version of the manuscript.

30. line 370 methylase gene associated with resistance to all aminoglycosides

Author: Line 444 of the revised version of the manuscript.

31. line 374 among all *A. baumannii* isolates examined, 13 different sequence types

Author: Line 449 of the revised version of the manuscript.

32. line 392 seen of which from site SA12 carried a bla oxa ...

Author: Line 470 of the revised version of the manuscript.

33. line 400 key for sites needs to be referenced

Author: This point is well received and key to the site and site number is added to all figures and tables whenever applicable.

34. line 411 are grouped together. Several isolates carried more than one

Author: Line 494 of the revised version of the manuscript.

35. line 425 with ESBLs included bla oxa

Author: Line 512 of the revised version of the manuscript.

36. line 427 at site BA 1 also harbored a bla NDM

Author: Line 514 of the revised version of the manuscript.

37. line 434 genes encoding for ESBLs

Author: Line 521 of the revised version of the manuscript.

38. line 446 isolates species include *K. pneumoniae* ...

Author: The sentence have been reformulated: "The two most frequently encountered bacterial species were *K. pneumoniae*... (Line 533 of the revised version of the manuscript).

39. line 460 56 different STs identified in 135 isolates

Author: Line 568 of the revised version of the manuscript.

40. line 462 and endemic in hospital settings

Author: The sentence has been reformulated (Line 584-586 of the revised version of the manuscript).

41. line 478 and were found to occur in a limited number of sites mostly linked with South Africa

Author: Line 591 of the revised version of the manuscript.

42. line 486 this study primarily relate to .. where lower numbers of patients were enrolled

Author: Line 598 of the revised version of the manuscript.

43. line 488 distribution of neonatal sepsis-associated pathogens

Author: Line 601 of the revised version of the manuscript.

44. line 497 showed a wide variety of bacterial species as causative agents of neonatal sepsis

Author: Line 615 of the revised version of the manuscript.

45. line 499 distributed among the sites studied (or sampled or monitored)

Author: Line 618 of the revised version of the manuscript.

46. line 500 second line WHO recommended treatments

Author: Line 619 of the revised version of the manuscript.